# Implementation outcomes of HIV self-testing in low- and middle- income countries: A scoping review

Adovich S. Rivera[1]*, Ralph Hernandez[2☯], Regiel Mag-usara[2☯], Karen Nicole Sy[2☯], Allan R. Ulitin[3], Linda C. O'Dwyer[4], Megan C. McHugh[1,5], Neil Jordan[6,7,8], Lisa R. Hirschhorn[9,10]

**1** Institute for Public Health and Medicine, Feinberg School of Medicine, Northwestern University, Chicago, Illinois, United States of America, **2** College of Medicine, University of the Philippines Manila, Manila, Philippines, **3** Institute of Health Policy and Development Studies, National Institutes for Health, Manila, Philippines, **4** Galter Health Sciences Library & Learning Center, Feinberg School of Medicine, Northwestern University, Chicago, Illinois, United States of America, **5** Department of Emergency Medicine, Feinberg School of Medicine, Northwestern University, Chicago, Illinois, United States of America, **6** Department of Preventive Medicine, Northwestern University Feinberg School of Medicine, Chicago, Illinois, United States of America, **7** Department of Psychiatry and Behavioral Sciences, Northwestern University Feinberg School of Medicine, Chicago, Illinois, United States of America, **8** Center of Innovation for Complex Chronic Healthcare, Hines VA Hospital, Hines, Illinois, United States of America, **9** Department of Medical Social Sciences, Feinberg School of Medicine, Northwestern University, Chicago, Illinois, United States of America, **10** Institute of Global Health, Feinberg School of Medicine, Northwestern University, Chicago, Illinois, United States of America

☯ These authors contributed equally to this work.
* adovichrivera2021@u.northwestern.edu

## Abstract

### Introduction

HIV self-testing (HIV-ST) is an effective means of improving HIV testing rates. Low- and middle-income countries (LMIC) are taking steps to include HIV-ST into their national HIV/AIDS programs but very few reviews have focused on implementation in LMIC. We performed a scoping review to describe and synthesize existing literature on implementation outcomes of HIV-ST in LMIC.

### Methods

We conducted a systematic search of Medline, Embase, Global Health, Web of Science, and Scopus, supplemented by searches in HIVST.org and other grey literature databases (done 23 September 2020) and included articles if they reported at least one of the following eight implementation outcomes: acceptability, appropriateness, adoption, feasibility, fidelity, cost, penetration, or sustainability. Both quantitative and qualitative results were extracted and synthesized in a narrative manner.

### Results and discussion

Most (75%) of the 206 included articles focused on implementation in Africa. HIV-ST was found to be acceptable and appropriate, perceived to be convenient and better at maintaining confidentiality than standard testing. The lack of counselling and linkage to care, however, was concerning to stakeholders. Peer and online distribution were found to be

**Funding:** Article processing fees were covered by the Northwestern University Open Access Fund.

**Competing interests:** The authors have declared that no competing interests exist.

effective in improving adoption. The high occurrence of user errors was a common feasibility issue reported by studies, although, diagnostic accuracy remained high. HIV-ST was associated with higher program costs but can still be cost-effective if kit prices remain low and HIV detection improves. Implementation fidelity was not always reported and there were very few studies on, penetration, and sustainability.

## Conclusions

Evidence supports the acceptability, appropriateness, and feasibility of HIV-ST in the LMIC context. Costs and user error rates are threats to successful implementation. Future research should address equity through measuring penetration and potential barriers to sustainability including distribution, cost, scale-up, and safety.

## Introduction

Human immunodeficiency virus self-testing (HIV-ST) is an effective means of improving HIV testing rates and can serve as an alternative to facility-based HIV testing [1, 2]. It is recognized as a key tool to help achieve the global 90-90-90 targets, specifically, ensuring that 90% of people with HIV are aware of their status [3, 4]. Many countries are in the process of integrating HIV-ST into their national HIV/AIDS programs [5]. This process can easily become complicated and poor implementation can lead to countries not gaining any benefits (e.g., address the HIV testing gap) from the new technology. Fortunately, implementation science has emerged in response to this problem of introducing new technology to health systems and health organizations have recognized that implementation itself requires a scientific approach and reviews of the current evidence are valuable in this approach.

One approach to organizing this review is according to implementation outcomes. Using Proctor et al.'s framework [6], implementation outcomes are different from the usual outcomes studied in clinical or health services research. Usually, outcomes in health research are patient outcomes (e.g., satisfaction or improvement in health and function) or quality of care outcomes (e.g., efficiency, safety, and timeliness) which are end products of an intervention. Meanwhile, implementation outcomes are indicators of success that are "the effects of deliberate and purposive actions to implement new treatments, practices, and services." Further, Proctor et al. proposes that only programs with successful implementation outcomes will achieve maximal effectiveness and quality. These eight outcomes are *acceptability*, *adoption*, *appropriateness*, *costs*, *feasibility*, *fidelity*, *penetration*, and *sustainability*. By identifying the relevant conditions necessary to achieve HIV-ST implementation outcomes, we can better understand how best to implement this technology for a given setting.

Another important factor in implementation is the context (i.e., where HIV-ST will be implemented). For example, the design of the national health system puts constraints on how HIV ST can be delivered to the end-user. The dominant modes of HIV transmission in the country are contextual factors that influences what populations will be prioritized by countries. Finally, the level of stigma faced by at-risk populations affects how they would react to HIV-ST. While there have been several reviews on HIV-ST implementation outcomes, these reviews primarily included studies conducted in high-income countries [7–11] with a couple of reviews focused on implementation in Africa [12, 13]. We agree that the reviews are very much useful. They found evidence that HIV-ST was acceptable to a wide variety of patients and providers and warn about issues that may hinder successful implementation like the high

cost of the kit, inadequate linkage to care, and the complexity of packet instructions. But there is a huge diversity of low- and middle-income countries (LMIC) contexts, findings and lessons for upper-income countries or from Africa may not always be applicable to other LMIC settings. Our review is important to guide individual country efforts in planning implementation research and programs of HIV-ST in their local settings.

In this paper, we aimed to describe the implementation outcomes of HIV-ST in LMIC using a scoping review methodology [14]. We then produced a synthesis of the existing literature to inform the design and implementation of HIV-ST programs. We also identified gaps in the literature to help the development of the research agenda for HIV-ST implementation. We hope this review serves as a resource for both researchers and practitioners as they study, or design programs aimed at effectively implementing HIV-ST in LMIC settings.

## Methods

### Protocol and search strategy

We conducted literature searches in Pubmed MEDLINE, Embase, Global Health, Web of Science, and Scopus. The final search strategy used combinations of the words "self-testing," "HIV," and "low- and middle-income country". The final strategy was informed by search strategies of previous reviews and a preliminary search (See S1 File for Sample Search Strategy). We also performed 1) hand search of references of reviews on self-testing and included articles, and 2) additional searches in potential gray literature repositories such as the WHO Library Database (WHOLIS), United States Agency for International Development data clearing house, Latin American & Caribbean Health Sciences Literature (LILACS), OpenGrey, and the study registry hosted by HIVST.org. We employed no language restrictions during the search and screened any article published from database inception to search date (25 January 2019). A protocol including the search strategy, inclusion criteria, and preliminary analyses plan was devised before performing the search (Version 1 at: https://doi.org/10.21985/n2-d8te-ey62). As with most scoping reviews, we iteratively revised the extraction and analysis plan as we reviewed the literature. The first extraction and analysis cycle was completed in April 2020. We performed an updated search on 23 September 2020 to cover new literature published since the end of the first analysis.

### Outcomes

We followed Proctor et al.'s definition of implementation outcomes [6] for this study with some modifications to facilitate categorization of studies (See S2 File for notes on definitions). During extraction, we recognized how of *safety* and *linkage to care* are intricately linked to successful implementation of HIV-ST, so we opted to include them in our review as additional outcomes. We did not expand the search for these two outcomes, rather, we focused on extracting only data on safety and linkage if a paper also reports one of the eight implementation outcomes.

One pair of outcomes that needed to be delineated operationally are *feasibility* and *fidelity*. Both outcomes relate to how well actual implementation aligns with the ideal implementation. We differentiate the two by separating the technology (HIV-ST) from the implementation program (i.e., how HIV-ST is reaching the end-user). We then treat *feasibility* as those related to successful use of HIV-ST as a piece of technology by the end-user. As such we included error rates and real-world diagnostic performance under feasibility. For *fidelity*, we focused on compliance of stakeholders to implementation strategies or described standard procedures employed within the HIV-ST program. Examples of these would be the procedure of giving two kits to each potential user or performing a follow-up call after self-testing.

Another pair that we had to differentiate was *adoption* from *penetration*. Both are related to the degree of uptake of technology in a population and both can be measured by determining how many among the eligible are using HIV-ST but occurs at different phases of implementation. Implementation can be viewed as a process with pre-implementation, early, middle, and late phases which can be described roughly as exploration, pilot testing, early roll out, and full institutionalization of new technologies [15]. We then interpret *adoption* studies to measure uptake in the early to middle phases as well as investigate if specific strategies lead to increased uptake by the target population. Meanwhile, we viewed *penetration* as a late-stage measure that requires the technology to be available for a longer period than in *adoption* and is quantified both service access and spread. Proctor's definition also has an added dimension of how well implementation strategies or processes have become part of usual care. Using this heuristic, we considered studies that looked at how many people have used HIV-ST before or during early implementation programs as under *adoption* and studies that measured uptake at later phases under *penetration*.

## Inclusion and exclusion criteria

We included any study on HIV-ST that reported at least one implementation outcome (*acceptability*, *appropriateness*, *adoption*, *feasibility*, *fidelity*, *implementation costs*, *penetration*, and *sustainability*) as defined by Proctor et al. [6] and was conducted in a World Bank classified LMIC. There were no restrictions on population or study design. We excluded (1) editorials, commentaries, or reviews, (2) articles written in a language other than English, (3) diagnostic accuracy studies that did not use results as read by the lay user in assessment, and (4) conference abstracts.

## Study selection and data extraction

We first screened articles based on their title and abstract and then performed a full-text assessment using Rayyan [16] and an Excel form. We extracted the following information from the included articles: first author's last name, publication year, study design, study population, data collection period, sample size, type of fluid for self-testing, and degree of supervision for self-testing. We extracted quantitative and qualitative findings for the different implementation outcomes using an Excel form that was iteratively revised. For qualitative findings, we focused on summary themes or statements. At least two authors were involved in each step of the selection and extraction. Conflicts were resolved by consensus formation.

## Analysis

We used descriptive statistics and exploratory data analysis to assess common design features across studies (ran in R 4.0). For each implementation outcome, we used narrative synthesis techniques [17] such as grouping (e.g., report quantitative results according to study design) and thematic analysis of reported qualitative themes to identify any similarities and differences in study results. We pooled diagnostic accuracy (sensitivity and specificity) results using bivariate modeling that was implemented through the *mada* package [18]. Sensitivity or specificity was extracted from the text or calculated from two-by-two tables. For calculations, we excluded indeterminate test results.

## Results

### Overview of results

The initial search yield was 1,131 unique (i.e., deduplicated) articles from the databases and 240 from other sources. From this, 196 remained after the title and abstract screen, and 107

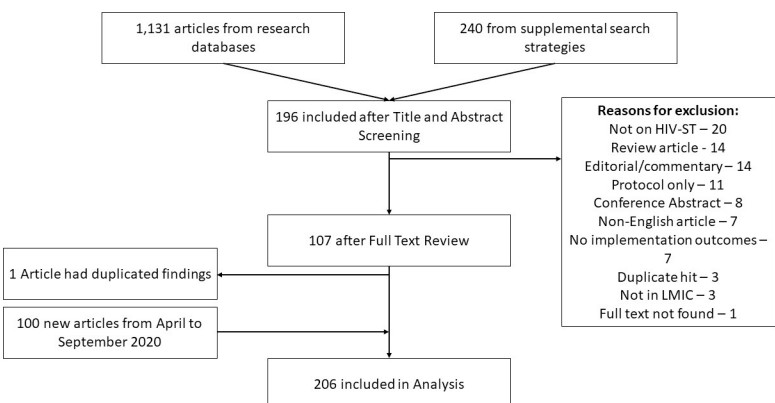

**Fig 1. Flowchart of search.** Notes: HIV-ST–HIV self-testing, LMIC–low- and middle-income country.

remained after the full-text review. Two included articles [19, 20] had very similar results and were treated as a single article. The published version of one article could not be found so the thesis version [21] was used instead. In the updated search, we found 100 additional articles leading to a total of 206 articles published between 2006 to September 2020 that were included in the synthesis. (Fig 1, See S2 File for List of Excluded Studies during Full-Text Screen).

The most studied outcomes were *acceptability* (59% of articles), *adoption* (50%), and *appropriateness* (27%). There were several studies on *feasibility* (12%), *fidelity* (11%), and *cost* (8%) but only two for *penetration* and one for *sustainability*. *Safety* (19%) and *linkage to care* (29%), the two additional outcomes, were also commonly discussed (Table 1).

The majority of studies were observational in design (72%), used quantitative methods only (63%), and were recently published (2019: 23%, 2020: 27%). Oral fluid-based (48%) and unsupervised (41%) self-testing were more commonly studied, although several studies failed to explicitly state these details. Most of the studies were on implementation in African contexts (70%) and were situated in four countries: Kenya (33, 16%), South Africa (33, 16%), Malawi (28, 14%), and China (27, 13%) (Fig 2). Nearly all papers studying implementation in Asia and the Americas involved men who have sex with men (MSM) while there was more variety in study populations among those in Africa. Most randomized trials were done in African contexts. (Table 1) (See S2 File for Detailed Study Descriptions and Extraction).

## Acceptability

122 articles studied *acceptability*. "Willingness to use" was the most common way of measuring acceptability and other measures included ease of use, preference over standard tests, and willingness to pay. Sixty-six (66) used a quantitative design [22–87], 37 were qualitative [43, 88–123], and 19 were mixed or multi-method studies [20, 124–141]. Only the quantitative results are presented in this section; qualitative results are discussed with findings on *appropriateness* due to similarities in themes and results.

**Willingness to use and ease of use.** Most studies found that high proportions of (>70%) to their study population were willing to use HIV-ST but there was some variation across key populations. For example, willingness was consistently high among female sex workers (FSW) ranging from 72% [142] to 95% [30]. In contrast, rates in MSM were more varied, ranging from 51 [73] to 99% [77] (Fig 3). We noted that the two studies (conducted in Kenya [129] and in Brazil [73]) that reported low willingness to use rates included individuals who have never tested for HIV (self-test or usual testing). This inclusion may have led to the lower measure of

**Table 1. Summary characteristics of included studies.**

| | Overall (n = 206) | Africa (n = 155) | Americas (n = 11) | Asia (n = 40) | p-value |
|---|---|---|---|---|---|
| **Key Population (%)** | | | | | |
| MSM | 56 (27.2) | 11 (7.1) | 10 (90.9) | 35 (87.5) | <0.001 |
| FSW | 25 (12.1) | 21 (13.5) | 0 (0.0) | 4 (10.0) | 0.371 |
| HCW | 23 (11.2) | 20 (12.9) | 1 (9.1) | 2 (5.0) | 0.358 |
| Pregnant women | 19 (9.2) | 18 (11.6) | 0 (0.0) | 1 (2.5) | 0.115 |
| Transgender women | 9 (4.4) | 0 (0.0) | 2 (18.2) | 7 (17.5) | <0.001 |
| Teenagers | 46 (22.3) | 38 (24.5) | 0 (0.0) | 8 (20.0) | 0.156 |
| **Type of Specimen (%)** | | | | | <0.001 |
| Blood | 21 (10.2) | 9 (5.8) | 1 (9.1) | 11 (27.5) | |
| Oral | 99 (48.1) | 88 (56.8) | 5 (45.5) | 6 (15.0) | |
| Both | 27 (13.1) | 15 (9.7) | 2 (18.2) | 10 (25.0) | |
| Not Specified | 59 (28.6) | 43 (27.7) | 3 (27.3) | 13 (32.5) | |
| **Type of Supervision (%)** | | | | | 0.012 |
| Unsupervised | 84 (40.8) | 68 (43.9) | 4 (36.4) | 12 (30.0) | |
| Supervised | 14 (6.8) | 6 (3.9) | 0 (0.0) | 8 (20.0) | |
| Not Specified | 83 (40.3) | 60 (38.7) | 6 (54.5) | 17 (42.5) | |
| Both supervision types | 25 (12.1) | 21 (13.5) | 1 (9.1) | 3 (7.5) | |
| **Type of Methods (%)** | | | | | 0.155 |
| Mixed methods | 23 (11.2) | 21 (13.5) | 0 (0.0) | 2 (5.0) | |
| Qualitative only | 54 (26.2) | 44 (28.4) | 2 (18.2) | 8 (20.0) | |
| Quantitative only | 129 (62.6) | 90 (58.1) | 9 (81.8) | 30 (75.0) | |
| **Study Design (%)** | | | | | 0.546 |
| Modelling | 3 (1.5) | 3 (1.9) | 0 (0.0) | 0 (0.0) | |
| Observational | 149 (72.3) | 114 (73.5) | 9 (81.8) | 26 (65.0) | |
| Interventional | 54 (26.2) | 38 (24.5) | 2 (18.2) | 14 (35.0) | |
| a) Quasi-experimental | 32 (20.3) | 18 (16.2) | 2 (18.2) | 12 (33.3) | |
| b) Randomized trial | 22 (13.9) | 20 (18.0) | 0 (0.0) | 2 (5.6) | |
| **Implementation Outcome (%)** | | | | | |
| Acceptability | 122 (59.2) | 93 (60.0) | 10 (90.9) | 19 (47.5) | 0.032 |
| Adoption | 103 (50.0) | 73 (47.1) | 4 (36.4) | 26 (65.0) | 0.085 |
| Appropriateness | 56 (27.2) | 46 (29.7) | 1 (9.1) | 9 (22.5) | 0.253 |
| Cost | 16 (7.8) | 15 (9.7) | 0 (0.0) | 1 (2.5) | 0.195 |
| Feasibility | 24 (11.7) | 20 (12.9) | 1 (9.1) | 3 (7.5) | 0.614 |
| Fidelity | 23 (11.2) | 17 (11.0) | 1 (9.1) | 5 (12.5) | 0.939 |
| Penetration | 2 (1.0) | 2 (1.3) | 0 (0.0) | 0 (0.0) | 0.717 |
| Sustainment | 1 (0.5) | 1 (0.6) | 0 (0.0) | 0 (0.0) | 0.848 |
| Linkage to Care | 59 (28.6) | 38 (24.5) | 4 (36.4) | 17 (42.5) | 0.068 |
| Safety | 40 (19.4) | 32 (20.6) | 1 (9.1) | 7 (17.5) | 0.609 |

Notes: MSM–men who have sex with men, FSW–female sex worker, HCW–healthcare worker.

overall willingness. One study in South Africa [48] found that the odds of being willing to use HIV-ST was two times higher among ever testers compared to never testers. Although, we also found a study in Tanzania [86] that did not find significant differences in the willingness to use rates of ever and never testers.

Nearly all studies concluded that HIV-ST was perceived easy to use [23, 28, 30–32, 35, 38, 40, 42, 44, 46, 47, 49, 57, 61, 63, 68, 70, 74, 77, 80, 82, 83, 85, 127, 130–134, 141]. Most studies

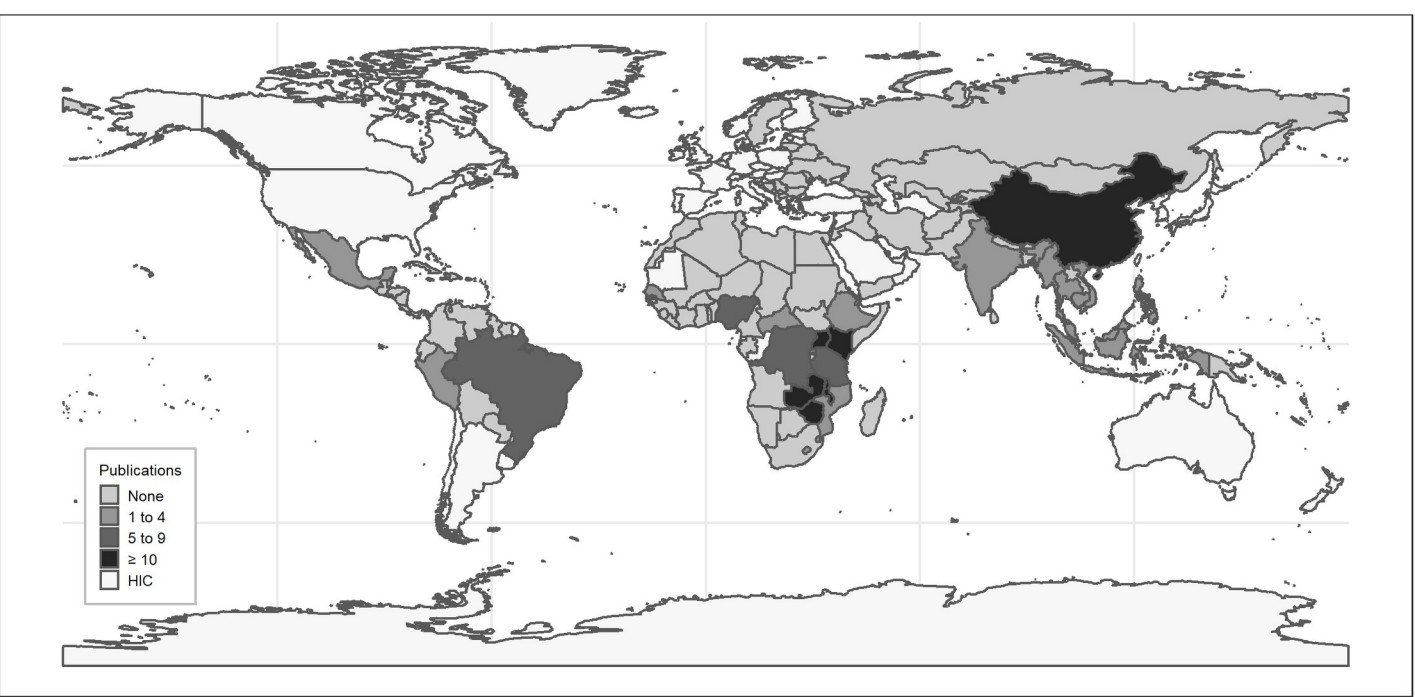

**Fig 2. Frequency distribution of included studies according to country.** Notes: HIC–high income country. Made with Natural Earth. Free vector and raster map data @ naturalearthdata.com.

also found that HIV-ST was preferred over standard testing (i.e. facility-based performed by a trained health care worker (HCW)) [22, 32, 39, 43, 46, 52, 61–63, 72, 80, 127, 130, 132, 136, 140].

**Willingness to pay.** Willingness to pay varied widely ranging from 21% [30] to 96.5% [133]. Most studies showed willingness greater than 50% across key populations. Some of the low rates are from populations with limited income such as students in Tanzania and Central African Republic [30, 75] and FSW primarily working in poor neighborhoods in Central African Republic [30]. The remaining study with low WTP [79] is on MSM in China where free HIV testing is available. Their sample also included a lot of people who have never tested which may affect WTP and the way they asked the WTP question ("Willing to purchase an oral self-test kit (in the next 6 months)?") seemed to capture both willingness to try and pay (Fig 3).

## Adoption

There were 103 studies on *adoption* included 18 randomized trials [25, 28, 47, 67, 143–156], 28 post-test only quasi-experiments [38, 43, 44, 49, 58, 61, 66, 68, 74, 79, 84, 85, 87, 127, 140, 141, 157–168], 30 observational quantitative studies [21, 29, 31, 32, 35, 37, 49, 59, 65, 77, 78, 81, 82, 148, 169–182], 21 qualitative studies [94–98, 100–102, 107, 113, 114, 119, 121, 183–190] and six observational mixed method studies [126, 129, 136, 138, 191, 192]. While *adoption* is distinct from *appropriateness*, we found that qualitative results for *adoption* overlapped with results for *appropriateness* and we opted to report them together (with *acceptability*) in the next section. We briefly go over observational studies in the next subsection and the rest is focused on quantitative results from interventional studies.

**Observational studies measuring uptake.** Observational studies showed that with just market availability and no widespread program by the State or national scale actors, HIV-ST

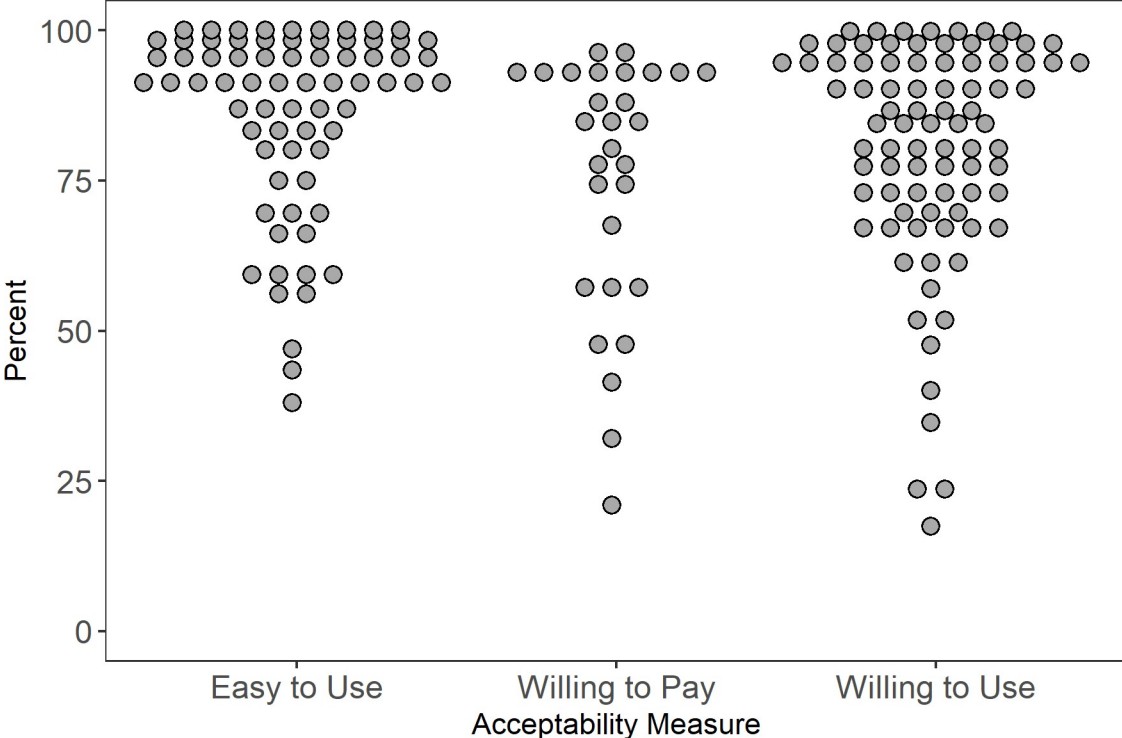

**Fig 3. Acceptability results.**

diffused among key populations. These surveys measured prevalent use of HIV-ST among HCW from Ethiopia [138], Kenya [21], and South Africa [31], among MSM in China [59, 170, 176, 178, 192], and the general population in Malawi and Zimbabwe [37]. Studies measured uptake either by asking about current use or reported uptake rates after offering the test kit for use during data collection. In the latter type, we found that those who adopt after being offered seem to represent a certain segment of the target population. For example, MSM in China who use gay apps were found to be more likely to use HIV-ST [193]. Women in community trial sites in Malawi [172] and Zambia [171] were more likely to participate and use HIV-ST. These two studies also showed significant differences in age and sexual behavior of HIV-ST users vs non-users. For example, in the Malawi trial, they found men, but not women, who recently practiced condom-less sex were significantly more likely to perform HIV-ST than those who did not.

**Overview of interventional studies.** Interventional studies have focused on evaluating the impact of implementation strategies on HIV-ST uptake or on overall HIV testing rates. The most commonly studied strategy was distribution of self-testing kits through community members, volunteers or peers [44, 61, 67, 74, 78, 85, 124, 125, 140, 146, 151, 159, 162, 194] or intimate partners (usually female) [28, 47, 49, 68, 144, 147, 152, 156, 161, 165, 166]. All except Tun et al. [74] used direct distribution of kits to users. Tun et al. [74], instead, asked peers in Nigeria to invite target users to visit a center to claim and use an HIV-ST kit. The other approaches for increasing uptake were distribution via online platforms [58, 66, 84, 87, 149, 157, 158, 160, 167], HIV-ST information or promotion campaigns [38, 67, 79, 150, 153, 155, 168], and offering HIV-ST after contact with health services (e.g., PreP, pharmacy visits, HIV testing) [25, 127, 141, 143, 145, 147, 154, 159, 164]. Many studies did not have controls that allowed for proper assessment of impact on HIV testing rates (Fig 4).

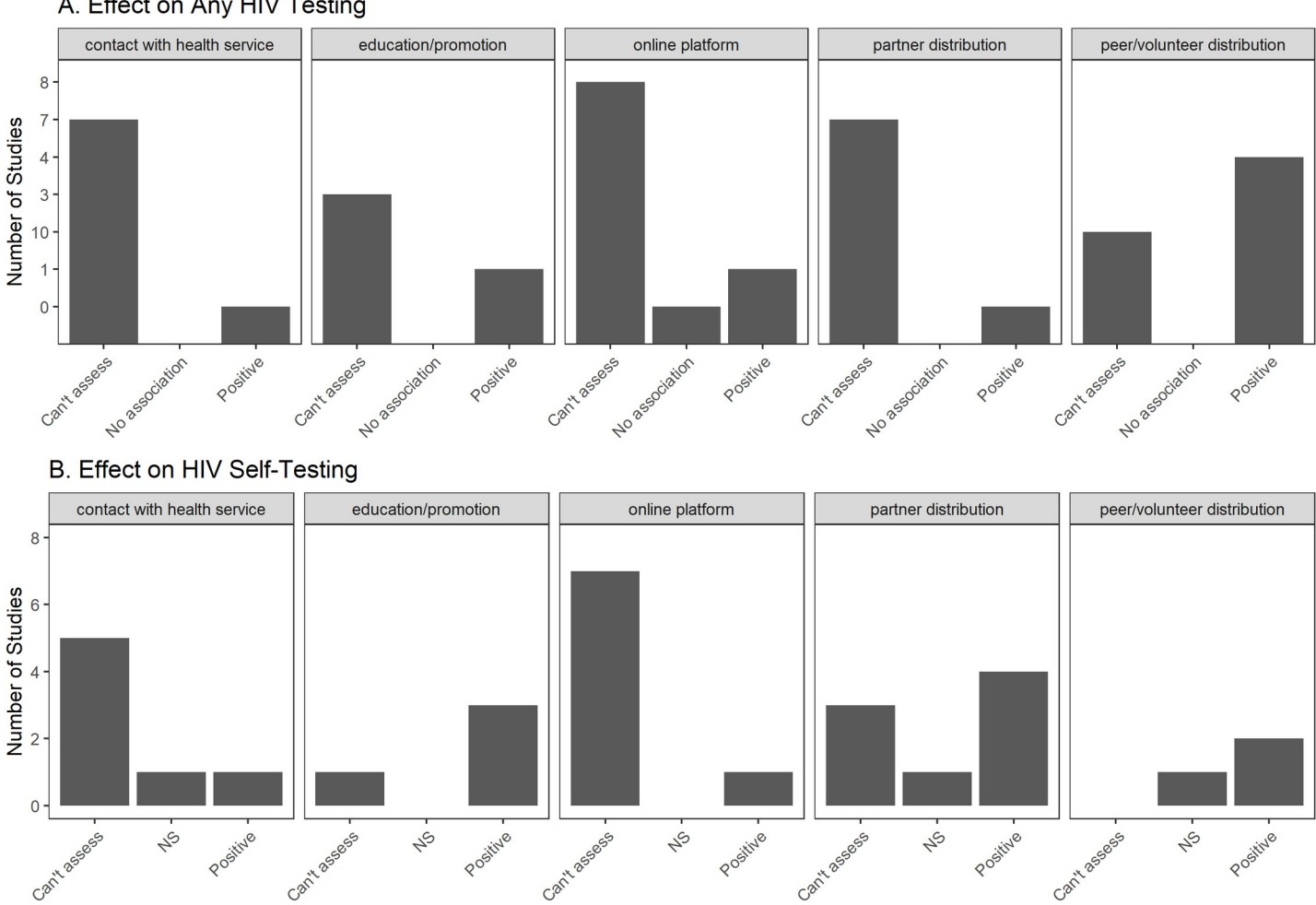

**Fig 4. Harvest map of interventional studies on HIV-ST adoption.**

**Distribution using peers, partners or community members.** The utilization of peers, intimate partners, or community members was mostly successful in increasing uptake. In quasi-experimental studies, there was high uptake ($\geq$80%) after direct distribution by peers [43, 78, 146, 151] or partners [28, 144, 152]. Nearly all randomized trials [28, 47, 124, 144, 146, 147, 156], except for one study in Zimbabwe [67], showed significant positive effects on overall HIV testing rates in the interventions arms. Also, Choko et al. [152] noted that in Malawi, distribution via partners with financial incentives can improve actual clinic follow-up by male partners.

The non-significant results in the Zimbabwe study [67] might be due to the comparison group. The Zimbabwe study had three factors in the intervention (price of kit, type of kit distributor, and promotional messaging). They found that needing to pay even just $0.50 significantly reduced uptake. In rural sites, they compared community health workers to retail stores as distributors and did not find any significant difference in uptake. They also found that type of messaging did not significantly affect uptake.

**Distribution via online platforms.** Online platforms were a promising approach to educate about and/or distribute HIV-ST, but only one of nine studies included control groups that enabled some form of evaluation on testing rates. Tang et al. [149] demonstrated through a

stepped wedge trial in China that a crowd-sourced online distribution coupled with a social media campaign can significantly increase the rate of HIV self-testing. Seven papers [66, 84, 87, 157, 158, 160, 167] showed that online distribution is feasible and led to the recruitment of HIV-ST users but no evaluation was done to assess performance compared to other distribution strategies.

A pertinent issue in online distribution is that HIV-ST users recruited online might not complete the counseling and testing process. One study in Thailand [58] demonstrated that a purely online approach to counseling and testing with HIV-ST had HIV testing completion rates comparable to the conventional in-person model. Another study in Brazil [87] also showed high rates of getting counseling among online users who tested positive.

**Other distribution strategies using education or promotion.**   Seven papers that used education or promotion campaigns through means other than peers or partners had mixed results. SMS announcements from HIV clinics in Kenya regarding HIV-ST availability significantly increased HIV testing among FSW [153] but not among truck drivers [155]. In China, inclusion in an SMS group that pushes HIV-related messages also increased HIV-ST uptake and overall HIV testing rates [150]. However, one study in Zimbabwe [67] found that merely adding promotional messages to voucher distribution did not improve HIV-ST uptake. Another study also found that a hospital-based information campaign among HCW in Kenya [38] led to high uptake among information session HCW attendees but not among non-attendees. Finally, two studies focused on comparing types of recruitment. One study in China [79] showed that online recruitment's impact on HIV-ST uptake was comparable to peer recruitment. Another China-based study [168] showed that social media key opinion leaders may be more effective than community-based organizations in recruiting HIV-ST users especially if they approach first-time HIV-ST users.

Some studies tested offering HIV-ST during contact with health services (e.g., routine visit, circumcision, pharmacy visits) and these had mixed results as well. Offering HIV-ST during a visit to an HIV testing facility seemed to increase HIV testing coverage. In Kenya, truckers who were randomized to the arm that can choose HIV-ST had 1.5x higher odds of accepting HIV testing compared to those randomized to just standard HIV tests [154]. Similarly, two other randomized trials in Malawi and South Africa [25, 147] that offered HIV-ST during facility visits or community events showed significantly higher HIV testing rates. A quasi-experimental study in Eswatini [164] showed an increase in HIV-ST uptake. Another quasi-experimental study done in Malawi, Zambia, and Zimbabwe [159] demonstrated the feasibility of distribution even in male circumcision sites, but they had no comparators for proper assessment. Integrating HIV-ST with pre-exposure prophylaxis (PreP) services also seemed helpful. In a sub-study of an open-label trial in Kenya [127], majority (93.2%) of recruited PreP users reported using HIV-ST at least once during the study. Offering HIV-ST during regular pharmacy visits in Kenya [141], however, did not have a high yield (only 35% of invited agreed to participate). Finally, in a community trial in Zambia [145], residents in communities that had access to HIV-ST (through community health workers) had higher odds of knowing their HIV status on follow-up than residents in control communities.

**Appropriateness and qualitative results.**   *Appropriateness* is defined as "the perceived fit, relevance, or compatibility of the innovation or evidence-based practice for a given practice setting, provider, or consumer; and/or perceived fit of the innovation to address a particular issue or problem" [6]. We found that all studies that touched on *appropriateness* used a qualitative design. While Proctor's framework differentiates between *appropriateness*, *acceptability*, *and adoption*, we found that qualitative themes for these outcomes relating to these three outcomes overlapped so we opted to present them together in this section.

**Positive perceptions on HIV-ST.**   Qualitative studies echoed findings of surveys described previously, HIV-ST was perceived to be easy and convenient, often in comparison to standard testing [57, 78, 88, 91–93, 95, 97, 99, 100, 102, 103, 105, 107, 108, 111–114, 120, 122, 124, 127, 131, 133, 137–140, 183, 184, 191, 195–197]. HIV-ST allows them save time and money as they would not need to wait in a line at a testing facility, skip work, or spend on transportation [20, 57, 88, 92–94, 99, 100, 102, 107, 111, 113, 114, 116, 119, 122, 126, 131, 137, 140, 183, 184, 191, 196].

HIV-ST also had potential to reduce stigma and anxiety associated with testing [9, 20, 45, 91, 95, 98, 99, 103, 104, 107, 112, 114, 118–120, 122, 126, 127, 131, 138, 184, 198]. Compared to standard facility-based testing, HIV-ST was perceived to be better in protecting confidentiality [20, 57, 88, 91, 95, 98, 100, 103, 104, 108, 112–116, 118–120, 122, 124, 126, 127, 129, 131, 137–140, 183, 184, 195–197, 199] especially in settings where people doubt the ability of HCW to protect confidentiality [20, 57, 99, 100, 104, 113, 124, 126, 137]. It was a common belief that HIV-ST would be useful for increasing HIV testing rates in certain populations, such as men or young people [45, 48, 78, 90, 91, 95, 98–100, 102, 106, 116, 118, 120, 137, 139, 184, 190, 191, 196, 199].

Relatedly, HIV-ST was believed to empower users and promote autonomy [20, 91, 92, 95–100, 102, 104, 105, 118, 122, 124, 137–139, 184, 189, 190, 200, 201] since HIV-ST allowed the user to retain full control of their own health information. The self-test option may also address the feeling of coercion associated with standard testing since some patients felt that they could not refuse if asked to undergo HIV testing because they perceived nurses to be in a position of power [20, 91].

**Considerations for designing HIV-ST programs.**   Studies showed several considerations for the design and implementation of HIV-ST programs in LMIC settings to improve acceptability and appropriateness. Proper kit design for the target population was often mentioned. Design considerations included the instructional materials [90, 98, 100, 103, 107, 114, 120, 124, 131–133, 187, 190, 201–203] and the type of fluid (blood or oral fluid) [90, 92, 100, 102, 104, 106, 108, 110, 114, 116, 119, 124, 127, 131, 132, 137, 139, 184, 191, 196]. Instructions needed to be clear and easy to understand. Some papers suggest making sure that there are clear instructions on how to link to care or seek support and include a number that users can easily call. Other studies showed that videos or mobile apps can serve as supplemental forms of education materials. Cognitive interviews was shown to be very useful in helping identify issues with instruction materials and optimize prior to roll-out [203]. In terms of specimen type, blood might be more trusted than oral fluid-based tests in some settings or by some subgroups. However, users might also have more difficulty with performing blood-based tests, particularly in using the lancet.

Counselling, especially post-test counselling, was deemed necessary in HIV-ST programs [9, 20, 45, 78, 89–92, 98, 100, 101, 103, 106, 108, 109, 111, 113, 114, 116, 119, 120, 127, 129, 132, 137, 139, 140, 184, 190, 196, 201]. Relatedly, two commonly reported negative features of the HIV-ST: the lack of access to immediate counselling [45, 57, 91, 92, 98–101, 103, 104, 106–109, 113, 114, 116, 118–122, 124, 126, 127, 132, 135, 138–140, 184, 186, 190, 195, 197] and the lack of linkage to care (e.g., confirmatory testing, anti-retroviral therapy) [45, 89–91, 98–100, 103, 104, 106, 108, 109, 113, 116, 121, 128, 129, 140, 186, 190, 196]. These two issues were voiced more often in the context of positive HIV-ST results. A commonly reported belief was that individuals who test positive would be more likely to resort to self-harm in the absence of counselling or post-test support [45, 88, 89, 91, 93, 98–101, 106, 107, 109, 113, 114, 116, 120–122, 129, 131, 132, 139, 140, 186, 190, 195, 196]. Some studies, however, suggested that there was no need to use traditional face-to-face facility-based counselling; telephone or online

counselling could be sufficient [20, 57, 90, 91, 98, 106, 108, 111, 112, 119–121, 132, 184, 190, 201].

Selecting the appropriate mode of kit distribution was important to ensure accessibility and adoption [57, 78, 90, 92, 99, 104, 105, 108, 110, 112, 119, 122, 137, 139, 141, 163, 184, 189, 190, 202, 204]. Modes of distribution described include: (1) direct delivery by community health workers, peers, partners, and sex workers, (2) Home delivery, and (3) Pick up at health facilities or community-based organizations or other sites (e.g., health clubs, markets, private pharmacies, vending machines). Regardless of the mode, the venue for distribution needs to be accessible and convenient. Venues need to ensure that the user privacy is protected, and the discretion is maintained. Peer or community distributors need to be perceived as trustworthy individuals, preferably coming from the potential user's own network. Users also need to be informed about storage strategies, especially if their homes have limited spaces for private testing.

**Issues related to partner distribution.** Partner distribution is an often-used strategy for distributing HIV-ST kits. Several papers explored issues of acceptability and appropriateness [94, 95, 99–102, 105, 107, 114, 118, 119, 121, 122, 127, 183, 184, 186, 189, 191, 196] and all except one [121] were done in African countries. Female partners find this strategy acceptable because it allows them to engage their partners in matters of sexual health and see it as a good way to encourage HIV testing and even couple testing (testing at the same time). Trust and relationship quality play important roles in the perceived acceptability or success of this strategy. Some females feel that it can improve their relationship and trust in their partners, but some fear that it can have the opposite effect. Males might view even view the request to use HIV-ST as a test of their fidelity.

Relatedly, the safety of female distributors is an important consideration in program design. Female distributors were believed to be at risk of physical violence, abandonment, and relationship dissolution. A qualitative study in Malawi [95] found that "women feared being branded as unfaithful and a possible domestic violence" if they bring up doing HIVST together. A similar study in Uganda also documented that women fear their partner's reaction the test although, in this study, none of the participants reported serious adverse events after offering the kit to male partners except for angry reactions among those who lied about the purpose of the test [102]. Results of quantitative studies on safety and social herms are reported in the safety section below.

Studies seem to agree that to mitigate these risks (e.g., negotiation skills training) must be included in the HIV-ST program if partner distribution is adopted as a strategy. A few papers [114, 183, 184, 186, 191] outlined strategies used by female distributors on how to engage their partners. Examples are: (1) selecting the appropriate time and place for introducing self- or couple testing, (2) developing a distraction free environment for the discussion and testing, (3) mentioning positive experiences and/or benefits of HIV-ST, and (4) reassuring partners that it is a couple's activity and not meant to test their commitment.

**Policies and other issues.** Modification of the policy environment to facilitate acceptance and adoption of HIV-ST seemed to be necessary for proper HIV-ST implementation. Policy issues included regulation or policies related to HIV-ST [45, 89, 97–100, 103, 108, 120, 123, 186, 190, 196, 200], pricing and financing [90, 98, 99, 103–105, 108, 118, 119, 123, 128, 131, 132, 137, 139, 187, 196, 202], and experience of frontline healthcare workers [141, 205]. There were calls for strict access to kits and putting in place protection against fake tests. One paper found that policy-makers from three African countries thought that policy should be "in line with WHO guidance and adapted to local conditions" [190]. Nearly all studies reported preference for free or heavily subsidized kits to ensure acceptability and adoption.

Relatedly, widespread availability of HIV-ST can also lead to coercive testing in the home and workplace and, in turn, threatens appropriateness. For example, parents can force their children to undergo an HIV-ST or a boss could require HIV testing of employees in the workplace or a spouse can coerce their partner during couples testing [94, 97, 100, 101, 116, 184, 190, 192].

Other reported considerations when designing HIV-ST programs included (1) adequate follow-up [100, 113, 118, 120, 127, 135], (2) establishment of peer support networks [78, 104, 107, 122, 140, 199] and, (3) proper education on HIV-ST use [93, 98–100, 102, 106, 108–110, 112, 116, 120, 122, 127, 131, 133, 139, 140, 190, 196, 201, 206]. Education, both at the individual and community level, ensures proper technique and addresses any concerns on test accuracy. Supervised testing may be more appropriate, especially for new users since they can provide additional education opportunities and emotional support. Despite overwhelming positive attitudes towards HIV-ST, a few studies [45, 100, 120] pointed out that HIV-ST will not improve testing coverage if unfavorable values towards HIV testing are not changed. Education to modify perceptions on HIV testing in general would then be needed. Education may also address potential problems like improper waste disposal [99, 119, 132, 190, 202], safety concerns (e.g., use of lancet as a weapon) [99, 137], psychological results of inaccurate tests [93, 190], sexual disinhibition due to negative results [100], and misconceptions regarding HIV or HIV-ST [90, 107, 110, 114, 116, 122, 132] (e.g., How can they trust oral testing if HIV cannot be transmitted via saliva?).

## Feasibility

As stated in the methods, for *feasibility*, we considered metrics that relate to the successful use of HIV-ST rather than compliance to program procedures or implementation strategies which we classified under *fidelity*. Metrics under this section include error rate, assistance rate, readability, and diagnostic performance.

**User errors and readability.**   Errors in performing HIV-ST were common. Studies reported error rates in different ways: some report the number of people with at least one error, others calculate an average or useability index, and some report error per step. Error rates of 10% or higher were commonly reported in studies that used unsupervised HIV-ST [24, 30, 40, 42, 46, 53, 57, 63, 77, 131, 132, 136, 142, 194] although error rates ranged widely from 1.6% to 93%. In these unsupervised HIV-ST studies, participants were observed but no guidance was provided outside of the education materials that were part of the kit simulating an unsupervised test. Meanwhile, supervised HIV-ST [23, 133, 173, 207] had lower error rates ranging from 0% to 19.5% (Fig 5). Gaining experience with HIV-ST did not lower error rates as seen in studies in South Africa, Brazil and Peru that compared error rates on the initial introduction and after months of unsupervised use [42, 77].

Requests for assistance was also reported [23, 30, 61, 127, 130, 133, 136, 142, 153, 155, 173, 208]. Studies on supervised HIV-ST reported higher assistance requests with rates ranging at 18.1% [130] and 41.5% [23] while studies on unsupervised ST ranged from 10% [136] to 23.6% [23] (Fig 5).

Seventeen studies performed readability assessments in which they asked participants to read one or more HIV-ST kit results but not necessarily their own [30, 32, 46, 53, 57, 70, 77, 80, 83, 133, 134, 136, 142, 173, 177, 194, 209]. Most studies reported >90% correct reader rate but this seemed to be dependent on the type of result (Fig 5). Inconclusive and weakly positive results commonly led to incorrect reading by the users.

**Test performance.**   Despite issues of performance and test result interpretation, HIV-ST, as used and read by study participants, was found to have good (albeit variable) diagnostic

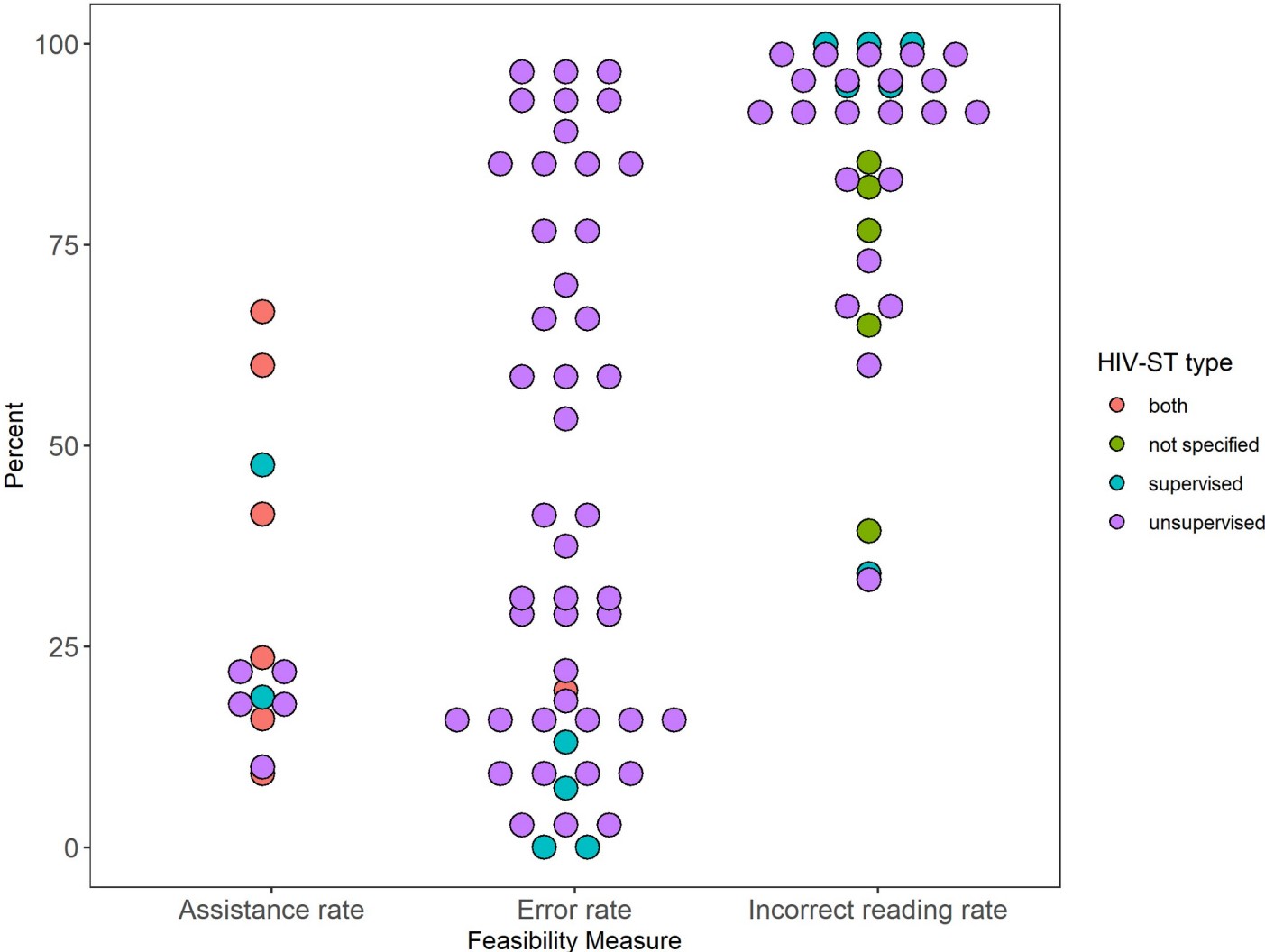

**Fig 5. Feasibility results.**

accuracy. To better summarize the results, we pooled results using meta-analysis methods for diagnostic studies. In 17 studies [23, 24, 32, 40, 57, 79, 85, 125, 133, 134, 136, 142, 174, 175, 177, 207, 210], sensitivity ranged from 66.7% to 100%, while specificity ranged from 81.3% to 100%. Pooled results from testing unsupervised oral HIV-ST and found that it is specific (98.7%, 95%CI: 97.6% to 99.3%) and fairly sensitive (90.6%, 95%CI: 84.9% to 94.3%) (Fig 6). User-readings usually agree with standard tests. Eight of ten studies with data [32, 132, 134, 136, 142, 175, 194, 207] reported high (≥90%) concordance between results as read by users compared to reference testing (e.g., done by a healthcare worker or using standard test). Similarly, seven of nine studies with kappa statistics showed high agreement between user readings and standard results [24, 30, 70, 133, 142, 175, 207, 209].

Finally, we found two studies that tested interventions to reduce user errors. A group in South Africa [211] developed a mobile app to aid HIV-ST users. While they reported only 8.7% user errors (at the low range compared to observational studies above), there was no control group to properly assess effectiveness. Another group performed a randomized trial involving Tanzanian youth [212] and they found that providing instructions using a graphic

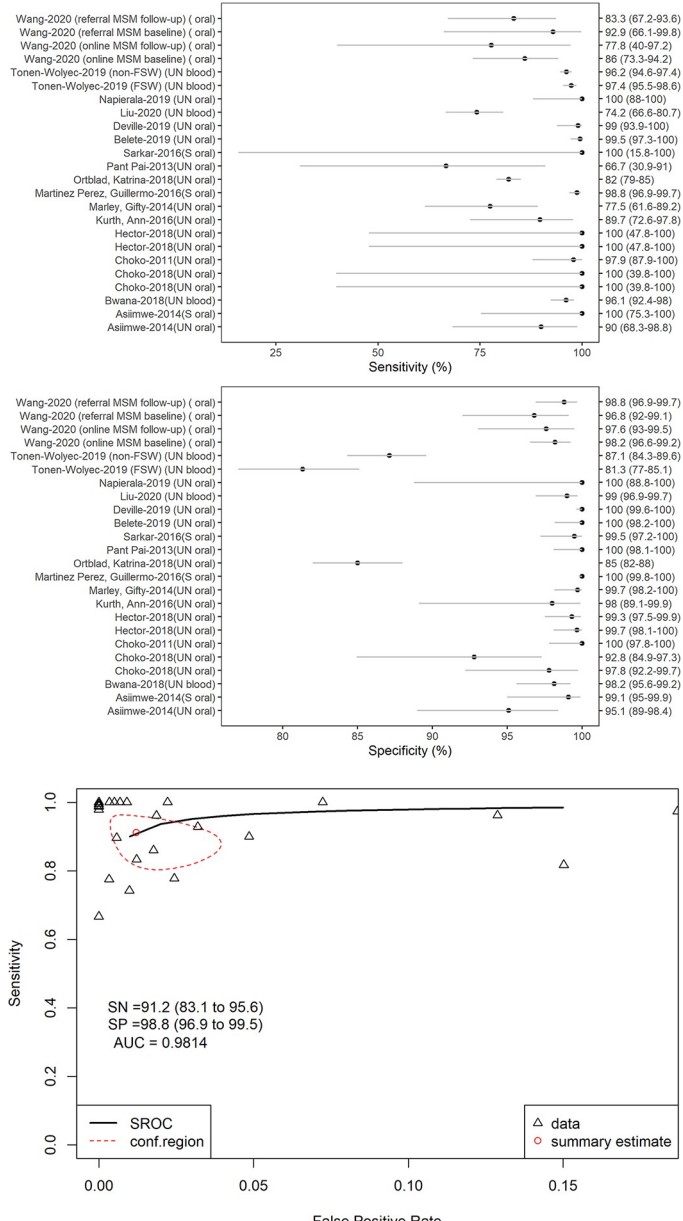

**Fig 6. Diagnostic accuracy of HIV-ST.** (A) Sensitivity (all studies), (B) Specificity (all studies), (C). Pooled Sensitivity and Specificity (Only studies using Unsupervised Oral HIV-ST). Notes: AUC–area under the curve, SN–sensitivity, SP–specificity, SROC–summary receiving operator characteristics, UN–unsupervised.

booklet and video was better than the graphic booklet alone. The booklet with video group had lower errors, better instruction comprehension, and higher intent to seek care.

## Implementation costs

Sixteen studies reported on *cost* in terms of implementation or cost-effectiveness. In African countries, offering HIV-ST generally increased total program costs or cost per person tested compared to standard HIV testing [25, 145, 204, 213–218] due to the higher cost of the testing kit and the added cost from strategies to promote adoption. For example, Mulubwa et al. [145]

noted a 1.37x higher cost in an HIV-ST program in Zambia partially due to added costs related to door-to-door distribution of HIV-ST kits. Higher costs were also observed by George et al. [216] in Kenya. They found that the HIV-ST treatment arms have double the cost of the usual care arms. The increase in program costs were due to the higher costs of HIV-ST kits (vs usual HIV testing) and the added costs of sending out SMS reminders to HIV-ST kit recipients.

While total costs may be higher, cost efficiency (e.g., cost per tested or cost per positive found) may improve with strategies that increase cost but also increase reach or yield. For example, Choko et al. [152] observed higher program costs in Malawi when HIV-ST was distributed through intimate partners. However, they found that while adding financial incentives would further increase program costs, the improvements in yield lowered the average cost per person who follows up for care. Another example is when Zhang et al. [168] showed that using social media key opinion leaders to recruit HIV-ST users in China yield lower cost per tested than using community-based organizations.

**Cost-effectiveness of HIV-ST.**   Despite higher program implementation costs, offering HIV-ST with standard testing can still be cost-effective (compared to standard testing alone). Maheswaran et al. [204] found that HIV-ST can be cost-effective in Malawi since it lowered societal cost by reducing patient opportunity costs. Cambiano et al. [219] found that adding HIV-ST can lead to cost-savings for the Zambian healthcare system. They also found that keeping the price of kits low ensured the cost-effectiveness of the program. Nichols et al. [218] found that aside from maintaining kit prices low, ensuring high linkage rates is also important to ensure that HIV-ST programs are cost-saving.

HIV-ST cost-effectiveness is also affected by the target population and HIV prevalence. Cambiano et al. [220] found that an HIV-ST program in sub-Saharan Africa is likely to be cost-effective for FSW and adult males but not for young people. Their results suggest that HIV-ST works best in communities where the prevalence is high and implementation costs can be maintained low. Korte et al. [156] echoes these findings in their CEA of distributing HIV-ST to males via their female partners in Uganda where they found that adding HIV-ST to existing testing programs would be best in areas in high prevalence.

The design and policies of the larger HIV program where HIV-ST is integrated also affect cost-effectiveness. For example, Maheswaran et al. [221] found that HIV-ST as a complementary service of standard testing can be cost-effective from a public provider perspective if positive individuals are managed following the 2015 WHO anti-retroviral treatment guidelines but not if the 2010 guidelines were used. Meanwhile, Johnson [222] showed that adding HIV-ST to offering a home-based HIV testing program in South Africa is cost-saving, but pairing HIV-ST with antenatal care partner testing programs is only cost-effective.

## Fidelity

Twenty-three studies reported some measure of implementation or program *fidelity*, defined as compliance to implementation strategies or operating procedures in the HIV-ST program. Even though we found several interventional studies, not all these papers reported fidelity measures. Since there were a variety of implementation strategies used, measures of fidelity also varied per study and depended on intervention type. Studies that used peer or partner distribution measured compliance of peer distributors to instructions [85], proportion receiving intended number of coupons [146], proportion of women or peers actually distributing self-tests [44, 47, 68, 125, 140, 147, 156, 165, 169, 171], or proportion doing couple-testing [47, 140, 156, 165, 166, 169]. Other examples of measures include: (1) number of visits by patients to the electronic platform [157], (2) number who completed testing after request via online platform [87, 160], (3) successful use of the testing support application [211], (4) compliance with

recommendation that was HIV-ST conducted upon opening a new pre-exposure prophylaxis bottle [127], (5) proportion of participants receiving phone call reminders [152], (6) number of SMS messages sent to participants [150], (7) reports of video issues during post-testing counselling [58], (8) adherence of counsellors to the protocol [205], and (9) using the kit after taking it home [61].

**Penetration and sustainability.** *Penetration* and *sustainability* are organization or setting-level constructs that refer to degree of institutionalization and maintenance of services. Both are late phase outcomes that require some implementation program to be in place prior to measurement. One measure of *penetration* is how many of the eligible users are using HIV-ST. Some studies that measured uptake could be considered as studies of this form of *penetration* [21, 37, 59, 82, 126, 170, 175, 176, 182, 192] with one study in Malawi using a more robust measure of population uptake by including questions in the national demographic health survey [37] rather than using convenience samples. We consider this as a passive type of penetration since HIV-ST uptake was not necessarily due to actual HIV-ST promotion programs and more attributable to market availability. We see in China (one of the countries where HIV-ST is widely available through e-commerce sites [223]), that the uptake rate of ever using HIV-ST can be at <20% [175] if measured outside of HIV-ST implementation programs.

There were also interventional studies that may affect *penetration*. We see from a cluster randomized trial of HIV-ST distribution by community health workers in Malawi [78] that community uptake takes some time but can reach high or maximal levels as long as the intervention is sustained. Further analysis of this trial's data [172] demonstrated, however, that high community uptake might be driven by people already reached by conventional and cheaper HIV testing services. The authors point out the need for demand creation activities so underserved groups would also be reached. Relatedly, a trial in Zambia [171] showed that secondary distribution with household distribution may be useful for targeting men and those older than 30 years old but tracking outcomes and minimizing costs would be challenging. Finally, a cohort study in China [180] showed that initiating HIV-ST might lead to increased engagement in facility-based testing which demonstrates HIV-ST's potential for improving the reach and long-term engagement with general HIV testing services.

We found only one study in Zimbabwe [125] that clearly described the organizational experience of routinization and scaling up of HIV-ST services and thus measures Proctor's *penetration* and *sustainability*. Scaling up of HIV-ST services in the program occurred only after three years of formative assessment and implementation. The program involves distribution of HIV-ST kits in 7 clinics and 12 outreach sites for FSW. The report showed that sustained implementation of HIV-ST. FSW were able to use the kits for themselves or distribute to their partners and clients. They were able to use routine data to monitor HIV-ST uptake including how many take the test on-site and how many agree to do secondary distribution. They also found that with adequate support, FSW were able to use HIV-ST accurately and did not encounter issues of linkage to care for that testing positive. The program was also able to maintain activities to routinely obtain feedback from FSW and the feedback subsequently informed their next steps.

## Linkage to care and safety

In addition to implementation outcomes, we found several studies that examined linkage to care rates (e.g., confirmatory testing, starting treatment) and decided to include these in the report. While studies often reported the number of people who tested positive, it was difficult to calculate linkage rates because they did not always delineate between those who tested

positive with HIV-ST, those who sought confirmatory testing, and those who had confirmed positives. Linkage to care also did not seem to follow a standard definition with some treating linkage as synonymous with starting ART and others treating linkage as those who sought further assessment after getting a positive test.

**Linkage to care.** There was a mix of approaches to measuring confirmatory testing. Most studies verified receipt of testing by either having all HIVST users undergo confirmatory testing as part of the protocol or by looking at clinic records with only a third (12/36, 33%) relying solely on report by study participants. From the studies with available counts, we found that rates of confirmatory testing were often high [23, 24, 29, 57, 61, 66, 68, 74, 77, 81, 84, 125, 133, 140, 145, 147, 156, 160, 162, 164, 168, 174, 178, 180, 207, 224, 225]. Only two studies [81, 144] reported low rates of confirmatory testing. One [144] was a trial of HIV-ST distribution through female partners in Kenya that showed only 25% (2/8 positives) of those with positive results in the HIV-ST intervention arm sought confirmatory testing compared to 75% (2/4 positives) confirmatory testing rate in the control arm. The other was a pilot trial of distribution of HIV-ST kits to male clients of brothels in Indonesia [81] where they detected two HIV positive results among the 188 who agreed to show their results but on follow-up they did not find evidence of receiving additional testing.

Similarly, most studies that reported linkage to care of confirmed positive cases found more than 50% linkage rates [23, 29, 57, 74, 77, 78, 84, 133, 144, 151, 152, 207] with only two reporting <20% linkage rates [81, 156]. One study with low rates [81] was the pilot study in Indonesia described above. The other was a trial of HIVST distribution via female partners in Uganda that reported low linkage of males who tested positive.[156] Most studies also reported 50% or higher rates ART initiation [24, 25, 29, 61, 84, 133, 140, 151, 152, 160, 162, 207, 224–226]; only two studies (in China [168] and in South Africa [174]), both of which were focused on assessing strategies to improve uptake among males, reported <30% initiation rates. Both studies had systems in place to provide counseling and follow-up to those who tested positive but still observed low rates.

Two interventional studies in Malawi and Zambia [152, 227] allowed comparing linkage rates of HIV-ST recipients to those under standard care. In both studies, linkage was lower in the HIV-ST arm confirming concerns raised previously described in the appropriateness section.

**Safety and coercive testing.** The occurrence of adverse events like coercive testing, self-harm after positive test results, and interpersonal violence seemed to be low across the studies that included these metrics [25, 47, 78, 85, 96, 97, 100, 125, 140, 145, 147, 150, 156, 162, 165, 169, 176, 178, 180, 181, 183, 188, 190, 192]. Some studies reported no serious adverse advents [78, 127, 144, 152]. Coercive testing was experienced by 2% to 9% of users with perpetrators being partners and healthcare workers and occurred via deception or threat of harm. While self-harm is a major concern based on qualitative studies, this does not seem to occur widely. An active community surveillance system in Malawi did not detect suicide due to HIV-ST [78] although a study in China reported that suicidal ideation and violence among positive testers was common [178]. Interpersonal violence (including sexual abuse) was an issue in distribution programs and was documented in qualitative and quantitative studies. Occurrence of this in HIV-ST distribution studies [47, 68, 78, 85, 156, 165, 169, 188] varied widely ranging from 0% to 10.5%. There were also other forms of harm reported in studies. In a pilot trial in Uganda [85], two of 19 distributors reported facing hostility from potential users. Marriage dissolution was raised by participants in a qualitative study in Malawi [95] but the report of the trial where the qualitative study was nested did not report occurrence of this event.[78] In a Zambian trial [145], there were 13 reported social harm events among 13,267 participants in the HIV-ST arm and these events included two cases of couple separation, three cases of

emotional distress, and one cases of threat of suicide. In trial in Uganda [146], there were only two reported events among 632 HIV-ST users involving one case of verbal abuse and one case of emotional distress from a false positive result.

## Discussion

In this scoping review, we found that the HIV-ST literature on implementation in LMIC remained largely focused on early implementation outcomes. HIV-ST was acceptable in a wide variety of LMIC and key populations. While it was perceived to be appropriate serving as a convenient alternative or add-on to standard testing since it was better at protecting confidentiality, the lack of counselling was troubling to potential users. It had high sensitivity and specificity, but the occurrence of user errors threatened diagnostic performance. Limited evidence supports the effectiveness of distribution via peers, intimate partners, community members, and online platforms in increasing adoption of HIV-ST. HIV-ST often increased program costs but may still be cost-effective if it improves HIV detection (Table 2).

Our key findings are aligned with findings of previous reviews that covered high income settings or have been limited to Africa [7, 10–13]. Prior reviews concluded that HIV-ST is generally acceptable especially if available at a low cost and that can be used by target populations with good accuracy and low error rates. Like our review, prior reviews also found that HIV-ST was viewed as a convenient alternative to facility-testing that improves testing confidentiality,

**Table 2. Summary of findings and gaps according to implementation outcomes.**

| Outcome | Key Findings | Gaps |
|---|---|---|
| *Acceptability* | Target users were likely to be willing to use and pay for HIV-ST, especially those with prior experience.<br>HIV-ST was often rated easy to use. | More studies on other priority populations such as transgender women. |
| *Appropriateness* | HIV-ST was as a convenient alternative to facility testing that also promoted autonomy and protected confidentiality.<br>HIV-ST programs need to include strong linkage and counselling mechanisms and must ensure an enabling policy environment.<br>HIV-ST kits need to be designed to ensure ease of use and compatibility with local preferences (e.g., specimen type). | Most studies looked at appropriateness at the user level but have not assessed appropriateness from the perspective of program implementers or policy makers. Appropriateness in terms of integration with existing programs is also understudied. |
| *Adoption (Uptake)* | HIV-ST can passively diffuse through target populations just from market availability.<br>Partner and peer distribution were effective means of improving adoption. Online distribution was feasible but needs to be evaluated for effectiveness. | There are very few robust evaluations of implementation strategies to promote adoption. |
| *Feasibility* | User errors were common but might not necessarily affect diagnostic testing accuracy.<br>HIV-ST was highly specific but only fairly sensitive. | There is a need to identify which user errors affect sensitivity and specificity, and what are effective strategies that will lower user errors in terms of performing the test and interpreting the results. |
| *Fidelity* | Most studies that measured fidelity reported high compliance to implementation strategies. | Fidelity needs to be reported consistently and interpreted in terms of impact on HIV-ST uptake and/or HIV-ST test diagnostic accuracy. |
| *Cost* | Addition of HIV-ST would increase program costs mainly due to the cost of the kit, but programs can still be cost-effective if adding HIV-ST improves detection and treatment rates of HIV positives. HIV-ST might be more cost-effective in high prevalence areas and as a targeted strategy for hard to reach populations. | There is a need for more costing and cost-effectiveness studies. Guidance on how to do costing for HIV-ST programs would be beneficial to the field. |
| *Penetration and sustainability* | We saw one case of a sustained and scaled up program that utilized phased implementation and consistently involved target users in the process. Reaching target users can be challenging and would likely need more than one distribution strategy. | There is a need for more organization or system-level studies on this outcome. Inclusion of HIV-ST use in representative samples should be considered to measure population-level penetration. Studies should pay special attention to impact of and strategies for integration of HIV-ST into existing HIV testing programs. |
| *Linkage to care and safety* | HIV-ST linkage rates were comparable to standard of care, but studies often reported low positivity rates or used strict linkage policies that would be difficult to replicate in actual practice. | There is a need for long-term and large-scale studies to assess impact on linkage to care and safety. |

reduces stigma, and supports autonomy. These reviews also pointed out counseling and linkage as important issues that need to be tackled in implementation. The reviews generally had little information on effectiveness of adoption strategies, linkage to care, and safety and our review found several studies that tacked these issues. Notably, we found promising evidence for partner or peer-based distribution strategies. We also found quantitative studies that found issues related to linkage and safety occur but are uncommon. Despite the low rates of harm or losing to follow-up, it is important for HIV-ST programs to include features to prevent or address these issues including a surveillance and quick response system to mitigate serious events like suicide. The concept of safety should not be limited to physical harm, rather, it should include emotional or economic harm such as coercive testing, distress from positive results, couple separation, or threats of economic abandonment. Finally, while findings of our review and the prior reviews generally converge regardless of geographic region, we observed that published work was still absent in several countries. We stress the need for local research to inform each country's policies and programs.

While there is a wealth of evidence on HIV-ST implementation outcomes, there are still some gaps that remain (Table 2). Acceptability and appropriateness studies need to expand to cover key stakeholders such as transgender women, policy makers, and frontline service providers. While we learned important program design considerations from qualitative studies, the issues of integration with existing HIV programs need to be explored more. Studying integration is important because most studies often added HIV-ST on to existing standard testing rather than a replacement to standard testing. Relatedly, papers examining the experience of healthcare workers would be important for designing workflows and allocating human resources. We also found that acceptability may differ across target populations. One finding was that first time HIV testers might be less likely to be willing to try HIV-ST compared to experienced HIV testers and, if confirmed by local research, tailored messaging and ancillary services to improve acceptability (and uptake) of first time HIV testers might be necessary.

We found a lot of interventional studies but there is a need to improve reporting of these studies to facilitate evidence assessment and future synthesis. Incomplete reporting of study details and adoption outcomes hindered our assessment of the effectiveness of distribution strategies. While papers measured error rates, there is no standard as to what errors are major and would affect diagnostic accuracy. Despite several studies with multi-component interventions, only a handful reported any fidelity or cost outcomes. In addition, most interventional studies have focused on evaluating the impact of distribution strategies on uptake or adoption. The impact of alternative counseling and linkage models (e.g., pure phone or online counselling) is understudied. There should also be more studies that investigate strategies to address user error rates.

Finally, more studies on late-phase outcomes like cost, penetration and sustainment are needed. More cost and cost-effectiveness studies are needed since costs typically determine the scope of a country's HIV-ST program. Organization-level studies reporting on penetration and sustainment especially on how HIV-ST is transformed into a routine service that is part of the HIV testing system are also needed. As HIV-ST programs are slowly integrated into existing HIV programs, the potential impact of reduced face-to-face encounters with HCW should be considered. These encounters can serve as means to deliver health promotion messages that may not be related to HIV but are still important to health. Losing these encounters may lead to unintended health consequences that may need to be considered especially in cost-effectiveness analyses using the societal perspective. Larger studies should investigate appropriate implementation strategies that will ensure linkage and safety, especially if programs rely on peer or partner distributors. All these studies should follow best practices in performing implementation science research [228].

Our review has several limitations. We limited our review to English-language articles; this led to exclusion of several foreign-language articles, most from China. As this was a scoping review, we also did not perform quality assessment, and more focused systematic reviews will be needed to answer specific implementation questions (e.g., effectiveness of implementation strategies on adoption or diagnostic performance of HIV-ST in the field) while also doing a thorough risk of bias assessment. While we utilized several gray literature sources, there could still be valuable studies we failed to capture such as internal technical reports for government health agencies. Lastly, we altered Proctor et al.'s definitions to facilitate extraction and synthesis. Specifically, we interpreted qualitative studies on acceptability, appropriateness, and adoption together, and we categorized compliance to using the HIV-ST under *feasibility* while compliance to implementation strategies under *fidelity*.

## Conclusions

The accumulation of evidence from LMICs offers a rich resource for guiding implementation of HIV-ST programs in these countries. Evidence shows that HIV-ST is perceived as an acceptable and useful tool to improve HIV testing coverage in LMIC contexts. There are important implementation issues like lack of counselling and linkage to care, the common occurrence of user errors, and the higher cost of HIV-ST programs that needs to be addressed to maximize HIV-ST uptake. While studies on early phase implementation outcomes are useful for designing HIV-ST programs, there is a need for more robust studies on *adoption*, *cost/cost-effectiveness*, *fidelity*, *penetration*, and *sustainability* to guide scale up and roll out.

## Supporting information

**S1 File. Search strategies.** Word document file containing sample search strategy for Pubmed MEDLINE.
(DOCX)

**S2 File. Database of studies.** Spreadsheet file containing extracted structured and quantitative data. Information per sheet are as follows: (1) Definitions and notes on implementation outcomes, (2) List of Excluded studies during full text screen and reasons, (3) Study characteristics of Included Studies, (4) Quantitative Acceptability results, (5) Quantitative Adoption results, (6) Quantitative Feasibility results, (7) Quantitative Linkage results.
(XLSX)

**S3 File. PRISMA-SR checklist for paper.** Word document containing accomplished PRISMA-SR checklist for submitted manuscript.
(DOCX)

## Acknowledgments

We thank Adela Mizrachi for providing feedback on earlier versions of this manuscript. We thank the reviewers for their constructive feedback.

## Author Contributions

**Conceptualization:** Adovich S. Rivera, Linda C. O'Dwyer, Megan C. McHugh, Neil Jordan.

**Data curation:** Adovich S. Rivera, Ralph Hernandez, Regiel Mag-usara, Karen Nicole Sy, Allan R. Ulitin, Linda C. O'Dwyer.

**Formal analysis:** Adovich S. Rivera, Ralph Hernandez, Regiel Mag-usara, Karen Nicole Sy, Allan R. Ulitin.

**Methodology:** Adovich S. Rivera, Linda C. O'Dwyer, Megan C. McHugh, Neil Jordan, Lisa R. Hirschhorn.

**Project administration:** Adovich S. Rivera.

**Software:** Adovich S. Rivera, Linda C. O'Dwyer.

**Supervision:** Adovich S. Rivera, Linda C. O'Dwyer, Megan C. McHugh, Neil Jordan, Lisa R. Hirschhorn.

**Validation:** Adovich S. Rivera, Ralph Hernandez, Regiel Mag-usara, Karen Nicole Sy, Allan R. Ulitin.

**Visualization:** Adovich S. Rivera.

**Writing – original draft:** Adovich S. Rivera.

**Writing – review & editing:** Ralph Hernandez, Regiel Mag-usara, Karen Nicole Sy, Allan R. Ulitin, Linda C. O'Dwyer, Megan C. McHugh, Neil Jordan, Lisa R. Hirschhorn.

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
