## [Decision Letter · Decision Letter 0]

19 Feb 2021

PONE-D-20-41102

Implementation Outcomes of HIV Self-Testing in Low- and Middle-Income Countries: A Scoping Review

PLOS ONE

Dear Dr. Rivera,

Thank you for submitting your manuscript to PLOS ONE. After careful consideration, we feel that it has merit but does not fully meet PLOS ONE’s publication criteria as it currently stands. Therefore, we invite you to submit a revised version of the manuscript that addresses the points raised during the review process.

We look forward to receiving your revised manuscript.

Kind regards,

Matthew Quaife

Academic Editor

PLOS ONE

Journal Requirements:

2. Please amend your list of authors on the manuscript to ensure that each author is linked to an affiliation. Authors’ affiliations should reflect the institution where the work was done (if authors moved subsequently, you can also list the new affiliation stating “current affiliation:….” as necessary).

3.  We note that Figure 2 in your submission contain map images which may be copyrighted. All PLOS content is published under the Creative Commons Attribution License (CC BY 4.0), which means that the manuscript, images, and Supporting Information files will be freely available online, and any third party is permitted to access, download, copy, distribute, and use these materials in any way, even commercially, with proper attribution. For these reasons, we cannot publish previously copyrighted maps or satellite images created using proprietary data, such as Google software (Google Maps, Street View, and Earth). For more information, see our copyright guidelines: http://journals.plos.org/plosone/s/licenses-and-copyright.

3.1.    You may seek permission from the original copyright holder of Figure 2 to publish the content specifically under the CC BY 4.0 license. 

3.2.    If you are unable to obtain permission from the original copyright holder to publish these figures under the CC BY 4.0 license or if the copyright holder’s requirements are incompatible with the CC BY 4.0 license, please either i) remove the figure or ii) supply a replacement figure that complies with the CC BY 4.0 license. Please check copyright information on all replacement figures and update the figure caption with source information. If applicable, please specify in the figure caption text when a figure is similar but not identical to the original image and is therefore for illustrative purposes only.

Reviewers' comments:

Reviewer's Responses to Questions

**Comments to the Author**

1. Is the manuscript technically sound, and do the data support the conclusions?

Reviewer #1: Yes

Reviewer #2: Yes

2. Has the statistical analysis been performed appropriately and rigorously? 

Reviewer #1: Yes

Reviewer #2: I Don't Know

3. Have the authors made all data underlying the findings in their manuscript fully available?

Reviewer #1: Yes

Reviewer #2: Yes

4. Is the manuscript presented in an intelligible fashion and written in standard English?

Reviewer #1: No

Reviewer #2: Yes

5. Review Comments to the Author

Reviewer #1: This paper has a good value on promoting HIV self-testing strategy, althouh there has been severral similar reviews on HIV self-testing. My suggestions are as follows:

1. The authors provided too many kinds of implementation outcomes. Some kinds of implementation could be integrated, such as Adoption and peneration. Or pls give clearer definition on these types. Currently, it's not easy to differentiate the difference.

2. Page 5, line 92-95. These sentences were awkward. Suggesting to rephrase them.

3. Page 8, line 147. "Willingness to pay". Is there variation on this across different key populations?

4. Page 15, line 290. Pls check this saying. According to my field expericen, this approach is acceptatble.

5. Page 18, line 363. Typo error. should be "nine".

6. Page 23, line 475. It was self-reported data, or confirmed with proof like photos.

Reviewer #2: Thank you for the opportunity to review your interesting research.

- introduction seems too succinct? expanding further on what is already known about the topic and how your research fills the gap might be helpful.

- line 120 - are these the absolute numbers of papers found or are they percentages?

- line 121 - consider using the term men who have sex with men instead

- line 142 - can you expand on why the willingness to use was consistently high among FSW, and also provide further information as to why there was marked variations for MSM?

- line 147-148 - similar to above, why was there such a marked variation from 32% to 97%? highlighting some key lessons here would be helpful

- line 184 - why was this the exception - what made this study different from the rest?

- line 203 (and elsewhere) - this paragraph is an example whereby having consistent formatting when referring to various papers would be more informative to the reader - as a minimum, you should state the setting (country) for each study so the reader understands the context of these statements.

- line 209 - typo "social media"

- line 228 - delete "ok but need to clean up"

- Results section (in general) could benefit from better use of subheadings. I got lost after a while.

- line 311 - this paragraph touches on an important topic of the potential for social harms. It would be helpful to expand this further with data from a few key studies including what was being measured, how it was done, and what did they find

- line 357 - I may have missed this - but how did you pool your results - more detail in methods please.

- line 403/404 - this sentence is awkward and has some typos in it

- line 412 - brackets around 2019

- line 429 - there is some sloppy use of punctuation and spaces. This is but one example. Please carefully proofread your manuscript.

- line 441 - please add citation.

- line 478 - what was the reason?

- line 480-1 - why so low? (i.e. <20% linkage)

- line 509 - can you expand on the comparison between your findings and the other reviews conducted in high-income countries. What was common? different?

- consider discussing the potential loss of benefits from contact with a health professional if only using self-testing, e.g. counselling, access to testing for other STIs, vaccinations, etc...

6. PLOS authors have the option to publish the peer review history of their article (what does this mean?). If published, this will include your full peer review and any attached files.

Reviewer #1: No

Reviewer #2: No

---

## [Author Response · Author response to Decision Letter 0]

19 Mar 2021

We appreciate the constructive feedback from the editorial team and the reviewers and have made the necessary changes to address these comments. We have edited the manuscript to follow style requirements and fixed the error in author affiliations. We extensively edited the Introduction and Discussion sections to reflect comments and new insights we gained as we were preparing this revised paper. We also added new details in the Results to paint a more complete picture of selected studies we highlighted in the text. Please review the attached Response to Reviewer comments file to see a detailed response to each comment.

We would also like to mention that Figure 2 was created in R using vector map data obtained from the public domain site, Natural Earth. As per the Natural Earth website, no permissions are needed to use their maps for any purpose. We, however, added the clause “Made with Natural Earth. Free vector and raster map data @ naturalearthdata.com” in the notes for Figure 2.

We hope that our manuscript now meets PLOS ONE standards and is sufficient for publication.

---

## [Decision Letter · Decision Letter 1]

7 Apr 2021

Implementation Outcomes of HIV Self-Testing in Low- and Middle-Income Countries: A Scoping Review

PONE-D-20-41102R1

Dear Dr. Rivera,

We’re pleased to inform you that your manuscript has been judged scientifically suitable for publication and will be formally accepted for publication once it meets all outstanding technical requirements.

Kind regards,

Matthew Quaife

Academic Editor

PLOS ONE

Additional Editor Comments (optional):

Reviewers' comments:

Reviewer's Responses to Questions

**Comments to the Author**

1. If the authors have adequately addressed your comments raised in a previous round of review and you feel that this manuscript is now acceptable for publication, you may indicate that here to bypass the “Comments to the Author” section, enter your conflict of interest statement in the “Confidential to Editor” section, and submit your "Accept" recommendation.

Reviewer #1: All comments have been addressed

Reviewer #2: All comments have been addressed

2. Is the manuscript technically sound, and do the data support the conclusions?

Reviewer #1: Yes

Reviewer #2: Yes

3. Has the statistical analysis been performed appropriately and rigorously? 

Reviewer #1: Yes

Reviewer #2: Yes

4. Have the authors made all data underlying the findings in their manuscript fully available?

Reviewer #1: No

Reviewer #2: Yes

5. Is the manuscript presented in an intelligible fashion and written in standard English?

Reviewer #1: Yes

Reviewer #2: Yes

6. Review Comments to the Author

Reviewer #1: (No Response)

Reviewer #2: Thank you for addressing my comments adequately. All the best in your ongoing research in this area.

7. PLOS authors have the option to publish the peer review history of their article (what does this mean?). If published, this will include your full peer review and any attached files.

Reviewer #1: No

Reviewer #2: No

---

## [Editor Report · Acceptance letter]

23 Apr 2021

PONE-D-20-41102R1 

Implementation Outcomes of HIV Self-Testing in Low- and Middle- Income Countries: A Scoping Review 

Dear Dr. Rivera:

I'm pleased to inform you that your manuscript has been deemed suitable for publication in PLOS ONE. Congratulations! Your manuscript is now with our production department. 

Kind regards, 

on behalf of

Dr. Matthew Quaife 

Academic Editor

PLOS ONE